# Exploring the Cognitive Neural Basis of Factuality in Abstractive Text Summarization Models: Interpretable Insights from EEG Signals

Zhejun Zhang [ID], Yingqi Zhu, Yubo Zheng [ID], Yingying Luo [ID], Hengyi Shao, Shaoting Guo [ID], Liang Dong [ID], Lin Zhang * and Lei Li *[ID]

School of Artificial Intelligence, Beijing University of Posts and Telecommunications, Beijing 100876, China; zhejun.zhang@bupt.edu.cn (Z.Z.); yingqizhu@bupt.edu.cn (Y.Z.); zyb@bupt.edu.cn (Y.Z.); luoyy@bupt.edu.cn (Y.L.); jsjjkfq740118@bupt.edu.cn (H.S.); guost@bupt.edu.cn (S.G.); dongliang@bupt.edu.cn (L.D.)

* Correspondence: zhanglin@bupt.edu.cn (L.Z.); leili@bupt.edu.cn (L.L.)

**Abstract:** (1) Background: Information overload challenges decision-making in the Industry 4.0 era. While Natural Language Processing (NLP), especially Automatic Text Summarization (ATS), offers solutions, issues with factual accuracy persist. This research bridges cognitive neuroscience and NLP, aiming to improve model interpretability. (2) Methods: This research examined four fact extraction techniques: dependency relation, named entity recognition, part-of-speech tagging, and TF-IDF, in order to explore their correlation with human EEG signals. Representational Similarity Analysis (RSA) was applied to gauge the relationship between language models and brain activity. (3) Results: Named entity recognition showed the highest sensitivity to EEG signals, marking the most significant differentiation between factual and non-factual words with a score of −0.99. The dependency relation followed with −0.90, while part-of-speech tagging and TF-IDF resulted in 0.07 and −0.52, respectively. Deep language models such as GloVe, BERT, and GPT-2 exhibited noticeable influences on RSA scores, highlighting the nuanced interplay between brain activity and these models. (4) Conclusions: Our findings emphasize the crucial role of named entity recognition and dependency relations in fact extraction and demonstrate the independent effects of different models and TOIs on RSA scores. These insights aim to refine algorithms to reflect human text processing better, thereby enhancing ATS models' factual integrity.

**Keywords:** natural language processing (NLP); abstractive summarization (ABS); factual extraction; electroencephalography (EEG); representational similarity analysis (RSA)

## 1. Introduction

In the era of Industry 4.0, information technology rapidly promotes industrial transformation but simultaneously leads to information overload, exposing people to vast amounts of textual information. The challenge of information overload intensifies when there is a need to quickly extract critical information from tedious text for decision-making, highlighting data quality and accuracy. Natural Language Processing (NLP), a fundamental branch of Artificial Intelligence (AI), offers a viable solution to the problem. Specifically, Automatic Text Summarization (ATS), an NLP technology, has been widely applied in news extraction, academic research, business reports, legal analysis, content recommendation, and various other fields [1,2]. In recent years, transformer architecture has gained significant popularity, leading to notable advancements in the fluency and readability of Abstractive Summarization (ABS) for language models such as GPT, BERT, and BART. However, certain limitations remain regarding the factual accuracy of summaries [3]. The chatbot ChatGPT cautions users that "ChatGPT may produce inaccurate information about people, places, or facts" beneath its conversation interface. Factual errors include misrepresenting or omitting

information, distortion of logical relationships, and erroneous interpretation of facts [4]. In domains with high demands for information accuracy, like healthcare, law, and finance, factual errors could lead to severe consequences.

In many NLP tasks, neural networks are the most advanced machine learning methods. However, their interpretability is frequently challenged [5]. Applying word vectors and lacking structured textual data inputs increase model opacity, rendering their outputs challenging to interpret [6]. Enhancing model interpretability is crucial as it facilitates comprehension of the functioning of "black box" models, thereby enabling their improvement [7]. ABS entails Natural Language Understanding (NLU) and Natural Language Generation (NLG), making it complex. Improved interpretability aids researchers in comprehending the process of summarization generation, encompassing aspects such as fact retrieval within the model, determination of dependency relations, and identification of potential erroneous entity replacements. The enhanced interpretability enables a targeted reduction in factual errors while preserving the integrity of the original text information to enhance the summary compression rate, thereby improving the summaries' accuracy, fluency, and readability [8].

In recent years, the intersection of computational modeling and cognitive neuroscience has garnered increasing attention. Relevant research contributes to understanding how language and information are processed in the human brain, offering novel perspectives and methodologies for enhancing and optimizing existing NLP models [9,10]. For instance, by emulating cognitive characteristics observed in human linguistics tasks, significant advancements have been achieved in named entity recognition and other related NLP tasks [7]. On the other hand, recent evidence suggests that differences in language models may not be as pronounced when neural model parameters are analyzed alongside EEG or brain data [11,12]. However, it is worth noting that no prior studies have explored text summarization from a cognitive neuroscience perspective. Given the significance of text summarization in facilitating quick information extraction and efficient knowledge management, as well as the current limitations regarding factual accuracy within existing models, this study aims to apply theories and methods from cognitive neuroscience to establish correlations between human brain activities and ABS model performance. It is important to note that this study is fundamentally exploratory in nature, without a predefined hypothesis. This approach is intended to enhance interpretability in text summarization and advance interdisciplinary research in these fields. Specifically, the main contributions of this study are:

1. This study pioneers using EEG signals from a cognitive neuroscience perspective to investigate factual issues of ABS, offering novel insights into the relationship between models and human brain activity in language tasks;

2. This study compares variations in EEG signals corresponding to factual word phrases and non-factual word groups obtained through different fact term extraction methods (dependency relation, named entity, part-of-speech tagging, and TF-IDF). It was found that the distinctions in EEG were most significant for factual and non-factual word groups using named entity recognition and dependency relations. This lays a foundation for integrating extraction methods to better simulate human text processing;

3. This study employs Representational Similarity Analysis (RSA) to compare the correlation between typical deep language models (GloVe, BERT, GPT-2) and human brain activities, revealing significant differences in RSA scores among different models and periods of human brain activity under specific conditions. These findings suggest potential adjustments to enhance the model's resemblance to the functioning of the human brain and facilitate a deeper understanding of mechanisms involved in language processing tasks within the human brain.

## 2. Related Works

### 2.1. Abstractive Text Summarization

The advancement of ATS technology facilitates the generation of concise and coherent summaries that effectively address information overload [13,14]. Since 2020, Transformer models and their variants, such as GPT, BERT, and BART, have demonstrated remarkable performance in NLP, particularly in generative tasks [3]. Unlike extractive summarization, which directly extracts information from the source text, abstractive summarization requires a deeper understanding of textual semantics and employs NLG algorithms to rephrase critical points [15]. This approach makes ABS more similar to human-generated summarization and significantly improves fluency and readability [16]. The complicated generation process, however, presents a prominent challenge: factual accuracy. During summary generation, errors in entity replacement or inaccuracies in logical relationships are expected. Such factual errors are challenging to detect or rectify using current statistical metrics [17].

Defining "what is a fact" is a crucial foundation for addressing the issue of factual accuracy, yet no unified definition for it currently exists. Before 2020, most studies regarded relational triples as the fundamental form of facts, using them to improve the factual awareness of language models and precision. However, comprehensive factual triples are not always extractable. Existing research often involves the introduction of numerous auxiliary virtual entities and additional triples, conversions that contribute to the complexity of predicting links for two or more "arcs" [18]. In recent years, broader dimensions of fact definitions have been proposed. Some researchers defined facts based on the keywords in the original text, such as the TF-IDF method [19]. This approach is essentially statistical, assessing the significance of a word in a document or corpus, enabling an intuitive identification of the text's theme. However, such statistical methods fail to comprehend the relations between identified words and others. Part-of-speech tagging has also been utilized for fact extraction, assigning specific functional labels to each word in a text, such as nouns, verbs, or adjectives [20]. Different parts of speech may represent various factual elements and tagging them helps identify their specific roles in sentences, which applies to texts of diverse types and styles. Nonetheless, this method might be overly simplistic, potentially overlooking more intricate semantic relationships. Entity recognition [17] has also been employed for fact extraction, discerning items with specific meanings from the text, and viewing entities as facts, proving valid in downstream tasks like text summarization and information retrieval with minimal cumulative error. However, entity recognition might neglect important information unrelated to entities. Some researchers employ dependency relations as a fact extraction method to uncover intricate relations between text lexemes. Dependency relations aid in understanding the primary structure of texts, transforming input sentences into labeled tuples, and extracting tuples associated with predicate lexemes [21]. However, these relations mainly capture inter-word relationships. In complex sentences, if the dependency path between two words is too lengthy or contains numerous nodes, the facts derived from these paths might fail to fully represent the sentence's intended meaning.

Understanding the characteristics of human linguistic cognition is of paramount importance. Some researchers have significantly enhanced the performance of ABS models by simulating features of human linguistic cognition [22]. When humans read and comprehend texts, they typically rely on various cues to discern which information is pivotal or significant. These cues often stem from individuals' expectations and knowledge background, bearing a certain degree of subjectivity, and might operate subconsciously, making them challenging to emulate directly through computational models. EEG signals during natural reading might encompass elusive information, such as sentence structures or specific named entities. Applying EEG signals from natural reading to selecting or designing fact extraction methods that align with human cognitive characteristics or guiding extraction methods' amalgamation hold significant implications for producing accurate and reliable text summaries.

### 2.2. Linguistic Cognition and Neural Network Modeling

ABS encompasses NLP and NLG. Therefore, understanding the cognitive mechanisms of language comprehension and generation in the human brain may offer valuable insights into the ABS research. Neural signals from the human brain bridge the gap between the information processes' mechanisms of cognition and models. EEG measures brain electrical activity by placing electrodes on the scalp, enabling the observation and analysis of real-time brain activity during specific tasks. With the capability to capture millisecond-level changes in brain activity, EEG boasts a high temporal resolution, making it apt for studying rapid cognitive processes such as language comprehension and generation. Although its spatial resolution is lower than functional Magnetic Resonance Imaging (fMRI), EEG can still monitor activity at multiple scalp locations and identify electrode positions related to specific tasks. As the brain comprehends text when the meaning of a word mismatches the overall sentence meaning, a neural potential peak often emerges, characterized by a negative voltage shift between 300 ms and 500 ms. This Event-Related Potential (ERP) is termed the N400 effect [23]. The N400 predominantly resides in the central and posterior regions of the brain, especially in the middle and rear sections of the temporal lobe. In contrast to semantics, when dealing with the processing of intricate syntactic structures, the brain manifests the P600 effect, a positive voltage shift that peaks approximately 600 ms post-stimulus [24]. Spatially, it is associated with the brain's left temporal and parietal regions. The N400 and P600 effects are pivotal tools in cognitive neuroscience for studying language processing, delineating EEG response patterns within specific time windows related to semantic and syntactic processing, respectively.

In recent years, the interdisciplinary research between computational models and cognitive neuroscience in NLP has garnered increasing attention. Researchers have sought to intertwine these two domains from multiple perspectives. For example, Lamprou et al. [10] and Ikhwantri et al. [9] have elucidated the language processing of neural networks from a neurolinguistic viewpoint and directed the training of neural networks based on the human brain's text processing mechanisms. The objectives of these studies encompass understanding the human brain's operational mechanisms and optimizing the performance of NLP models. In terms of model optimization inspired by cognitive neuroscience, Y. Chen et al. [7] introduced a controlled attention mechanism for named entity recognition, which exhibited exemplary performance across multiple datasets. Besides, Ren et al. [25] successfully integrated cognitive signals into neural network NLP models through the CogAlign method, with experimental results indicating its efficacy in enhancing model performance. To gain a deeper comprehension of how to map models onto human brain activity, Oseki ns Asahara [26] designed a method to obtain EEG signals from participants during natural reading of a specific corpus and used the processed EEG signals to annotate various levels of the corpus. Oota et al. [27] further discerned that representations learned from different NLP tasks respectively interpret the brain responses to speech reading and listening: representations from semantic tasks (such as paraphrase generation, text summarization, and natural language inference) are more pertinent for listening comprehension, while those from syntactic tasks (such as coreference resolution and shallow syntactic parsing) are more pertinent for reading comprehension. Additionally, some researchers have ventured from cross-modal and multilingual perspectives. For instance, leveraging cross-modal transfer learning, Antonello et al. [28] discovered a low-dimensional structure that seamlessly bridges various linguistic representations learned by different language models, including word embeddings and tasks related to syntactic and semantic processing. This low-dimensional representation embedding reflects the hierarchical structure of language processing and can predict fMRI responses elicited by linguistic stimuli. On the other hand, Giorgi et al. [29], approaching from a developmental neuroscience perspective, introduced a neural network architecture designed to learn multiple languages concurrently.

RSA is a prevalent technique used to evaluate the relationship between deep language models and neural activity [30]. Lenci et al. [31] highlighted that RSA is particularly suited for datasets that are challenging to compare directly, such as neural brain activity and

internal representations of machine learning models. The core concept of RSA involves transforming raw data or model representations into a common similarity space, typically achieved by computing similarity matrices. The application of RSA unveils which sections of the neural network model are most similar to brain neural signals. Moreover, it can also facilitate cross-validation between brain data and multiple computational models to determine which model best accounts for brain activity.

In summary, integrating cognitive neuroscience and computational models allows for a deeper understanding of the linguistic processing mechanisms within the human brain. It facilitates the optimization and enhancement of existing NLP models. Research in this interdisciplinary field paves new avenues for future endeavors in NLP and neuroscience. However, despite various NLP tasks covered in this cross-disciplinary research, such as named entity recognition and natural language inference, there remains an absence of studies specifically applying insights from cognitive neuroscience to elucidate or optimize text summarization tasks.

## 3. Materials and Methods

### 3.1. Participants

This study recruited 14 valid participants, 9 males and 5 females, with an average age of 22.64 ± 2.90 years. All participants met the following criteria: (a) they had passed the English CET6 examination; (b) they had not dyed their hair in the past two months; (c) they had normal or corrected to normal vision, with no visual impairments such as color blindness or color weakness; (d) they had no history of psychiatric illnesses or mental disorders and no language or motor impairments; (e) they had not experienced physical discomfort (e.g., cold) in the past week and ensured adequate rest the day before the experiment.

### 3.2. Apparatus and Materials

This study used the BEATS system to collect 32-channel EEG data [32], and utilized PsychoPy (version 2022.2.5) for designing a continuous text reading task, capturing both stimulus timing and participant responses [33,34].

The experimental materials for this study were selected from the Factuality Evaluation Benchmark (FRANK) dataset [4]. This dataset comprises 2250 summaries generated by 9 models on 2 datasets (CNN/DM and XSum) and manual annotations. Each summary within this dataset has been meticulously annotated for factual errors, utilizing a detailed factual error classification system.

The criteria for material selection in this study included: (a) Presence of Factual Errors: Materials were required to have annotations indicating factual errors in model outputs; (b) Participant Fatigue Prevention: Each article was limited to a maximum of 200 words, with no more than 7 articles used per experiment, keeping sessions under 1 h to minimize participant fatigue; (c) Simplicity in Language: The chosen texts minimized the use of infrequent vocabulary and proper nouns, ensuring clarity in language processing tasks. The details of the selected materials are presented in Table 1. A total of 7 English short texts were chosen for the study. After merging all texts and tokenizing using the Python spaCy, 1220 tokens were obtained.

**Table 1.** Selection of experimental materials.

| Article Hash | Count of Sentences | Count of Words | Count of Tokens |
|---|---|---|---|
| 32143053 | 10 | 154 | 180 |
| 38329319 | 11 | 200 | 242 |
| 32457391 | 7 | 152 | 176 |
| 33652722 | 6 | 135 | 148 |
| 31566848 | 6 | 128 | 143 |
| 31920236 | 7 | 139 | 160 |
| 38595401 | 6 | 138 | 171 |

In subsequent sections, unless expressly stated otherwise, any mention of the term "word" refers to a "token."

### 3.3. Experimental Procedure

The experimental procedure could be divided into three stages: preparation, pre-experiment, and formal experiment. The study was conducted in a sealed laboratory, where the lights were turned off, and external noise was isolated during the formal experiment. In the preparation stage, the experimenter explained the experiment's purpose, process, and potential risks to the participants, ensuring that all participants read and voluntarily signed an informed consent form. Subsequently, the experimenter cleaned participants' scalps to ensure optimal electrode adhesion. Then, the experimenter fitted the EEG cap onto the participant's head and applied electrode gel to enhance the contact quality between the electrodes and the scalp. Lastly, the functionality of the EEG equipment and recording system was checked to ensure the correct setup of system parameters and that all electrode channels were connected correctly, thus ensuring the acquisition of the required data.

In the pre-experiment stage, participants followed a procedure similar to the formal experiment but only needed to read a shorter text to ensure they understood the experimental task. During the formal experiment, as illustrated in Figure 1, participants sat in front of a computer screen, gazing horizontally at it, and were instructed to minimize head movement throughout the session. Initially, a fixation cross appeared at the center of the screen for 5 s, followed by a pre-selected text presented word-by-word, with each word displayed for 1.5 s. Punctuation marks were not presented separately. After reading an entire text, participants were required to answer three comprehension questions to assess their understanding of the text. Only results from participants with a final accuracy rate above 50% were considered valid. Following the reading of each text, participants were given a break. If any discomfort arose during the experiment, participants could press the "Esc" key at any time to exit the experimental program. Throughout the experiment, one experimenter guided participants through the tasks while another monitored real-time EEG signals, saving the EEG data corresponding to each participant's reading of each text.

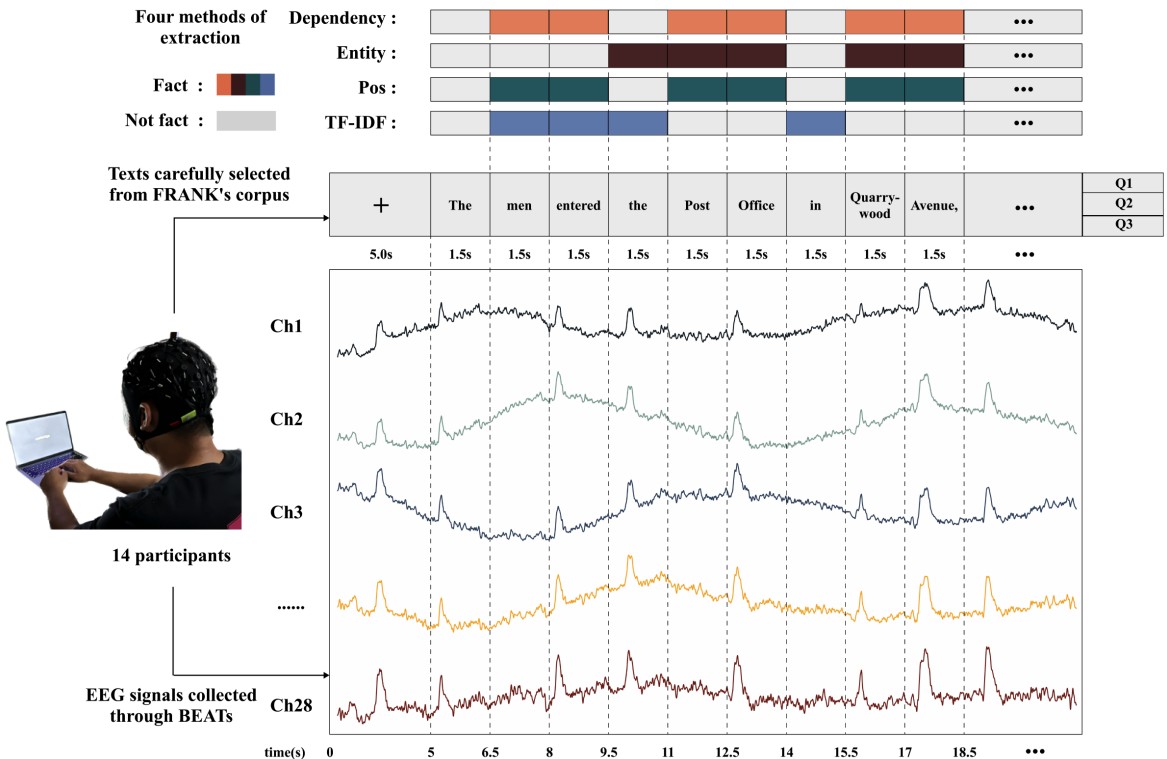

**Figure 1.** Experimental procedure.

### 3.4. Collection and Preprocessing of EEG Data

The EEG signals in this study were collected from 28 electrodes embedded in an elastic cap arranged according to the 10/20 system, as illustrated in Figure 2. Throughout the process of collecting, the impedance of each electrode was maintained below 10 kΩ, and the sampling rate for the EEG signals was set at 250 Hz. Preprocessing was conducted using Python's SciPy. During the preprocessing phase, the low-pass filter of EEG data had a cutoff frequency set at 30 Hz, while the high-pass filter's cutoff was set at 0.1 Hz. Filtering was executed independently on each channel. Subsequently, a reference transformation was applied to the EEG data to eliminate common noise and background signals across channels. This step involved calculating the average value across all channels at each time point and subtracting this average from each channel's signal. Finally, baseline correction was executed by determining the mean value for every channel over the entire recording duration and subtracting this mean from each respective time point within that channel.

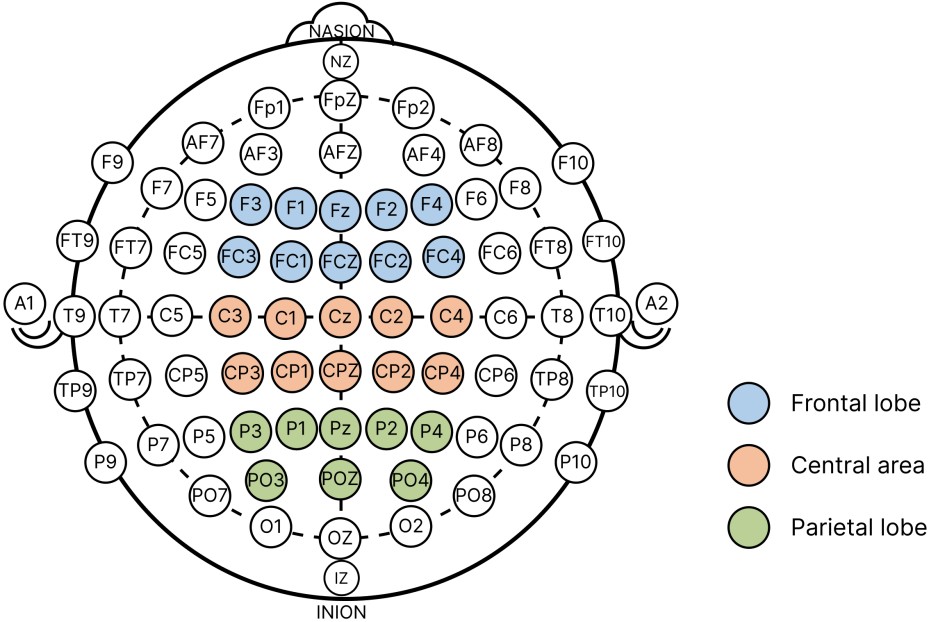

**Figure 2.** EEG channels selected in the study.

### 3.5. Metrics Selection and Data Analysis

3.5.1. Analysis of EEG Sensitivity to Different Fact-Extraction Methods

In the current study, Python's spaCy library was employed for preprocessing the corpus. As discussed in related works, four extraction methods (dependency, entity, pos, and TF-IDF) were selected. For clarity, any subsequent mention of the term "word" in this context refers to a "token."

For dependency relation extraction (dependency), spaCy initially identified dependency relations among all words. Then it selected labels representing relationships between words, including: 'ROOT', 'nsubj', 'nsubjpass', 'compound', 'poss', 'pcomp', 'ccomp', 'conj', 'relcl', 'dobj', 'pobj', 'iobj', 'appos', and 'acl', totaling 14 labels. These labels were considered capable of extracting fact-related information. As for named entity recognition (entity), spaCy tagged proper nouns in the text, such as names of people, places, organizations, and dates. Part-of-speech tagging (pos) was also executed using spaCy, extracting nouns and verbs from the text. When employing the TF-IDF method for extraction, Term Frequency (TF) and Inverse Document Frequency (IDF) were primarily calculated and then multiplied, as presented in the subsequent Formulas (1)–(3). The documents selected for IDF calculation included all entries from the FRANK dataset with word counts below 1000. When calculating TF, all words were considered equally important. However, common words like "the" and "a" might have appeared frequently but might not have been crucial. Thus,

we needed to downplay the weight of these words. IDF served as a method to reduce the weight of these words.

$$TF(t) = \frac{\textit{Number of times term t appears in a document}}{\textit{Total number of terms in the document}} \quad (1)$$

$$IDF(t) = \log\left(\frac{\textit{Total number of documents}}{\textit{Number of documents with termt in it}}\right) \quad (2)$$

$$\textit{TF-IDF}(t) = TF(t) \times IDF(t) \quad (3)$$

Cosine similarity can be employed to assess the similarity between vectors. Researchers utilize the cosine similarity of word weight vectors to compare the degree of similarity between documents or the cosine similarity of word vectors to compute the similarity between different words [35]. Inspired by prior studies, this research considers each token's corresponding EEG signal as an EEG vector, using their cosine similarity to assess the sensitivity of EEG signals to different factual word extraction methods. Specifically, after preprocessing, the EEG data from reading the text in this study was saved with a structure of $(n, 28, 375)$. $n$ was the number of tokens of the corresponding text after tokenization by spaCy. Here, 28 represented the number of channels for EEG signal collection, and 375 was the number of potentials collected within 1.5 s for each word. After averaging across the 28 channels, the data structure becomes $(n, 375)$, suggesting that each token corresponds to a 375-dimensional EEG vector. In addition to the full 1.5 s, this study also established 2 time of interest (TOI) intervals: the 250–500 ms interval, which might contain the N400 ERP component, and the 500–1000 ms interval, which might contain the P600 ERP component.

The cosine similarity between the EEG vectors of factual and non-factual word groups is calculated as presented in Equation (4). Here, $c_1$ and $c_2$ represent the centroids of the vectors for the factual and non-factual word groups, respectively. The centroids are determined by taking the mean of all vectors within the factual and non-factual word groups. Unlike the similarity between word vectors, which is always non-negative, the similarity range for EEG vectors lies between $-1$ and 1. A smaller angle between vectors indicates a closer directionality, making the cosine similarity approach 1. Conversely, a larger angle suggests divergent directions, bringing the cosine similarity closer to $-1$. When the angle is 90 degrees, the cosine similarity is 0, signifying that the vectors are orthogonal and unrelated. The method for calculating the cosine similarity between EEG vectors within either the factual or non-factual word groups is depicted in Equation (5). In this Equation, n is the total number of words within the word group, while $v_1$ and $v_2$ are the EEG vectors for the i-th and j-th words, respectively.

$$\textit{Inter-class Cosine Similarity} = \frac{c_1 \cdot c_2}{\|c_1\|_2 \times \|c_2\|_2} \quad (4)$$

$$\textit{Intra-class Cosine Similarity} = \frac{2}{n(n-1)} \sum_{i=1}^{n} \sum_{j=i+1}^{n} \frac{v_i \cdot v_j}{\|v_i\|_2 \times \|v_j\|_2} \quad (5)$$

3.5.2. Analysis of Correlation between Human Brain and Models

This study employed RSA to assess the similarity between model and human brain activity. The core concept of RSA is that when representational systems (e.g., neural network models or the human brain) receive many inputs, they measure the activity patterns generated by each input. By calculating the response similarity for each possible pair of inputs, one can construct a representational similarity matrix, which encapsulates the internal structure of the representational system. In this research, with 1220 tokens in the text, the analysis of the representational similarity between the model and human brain activity was divided into 3 steps, as depicted in Figure 3.

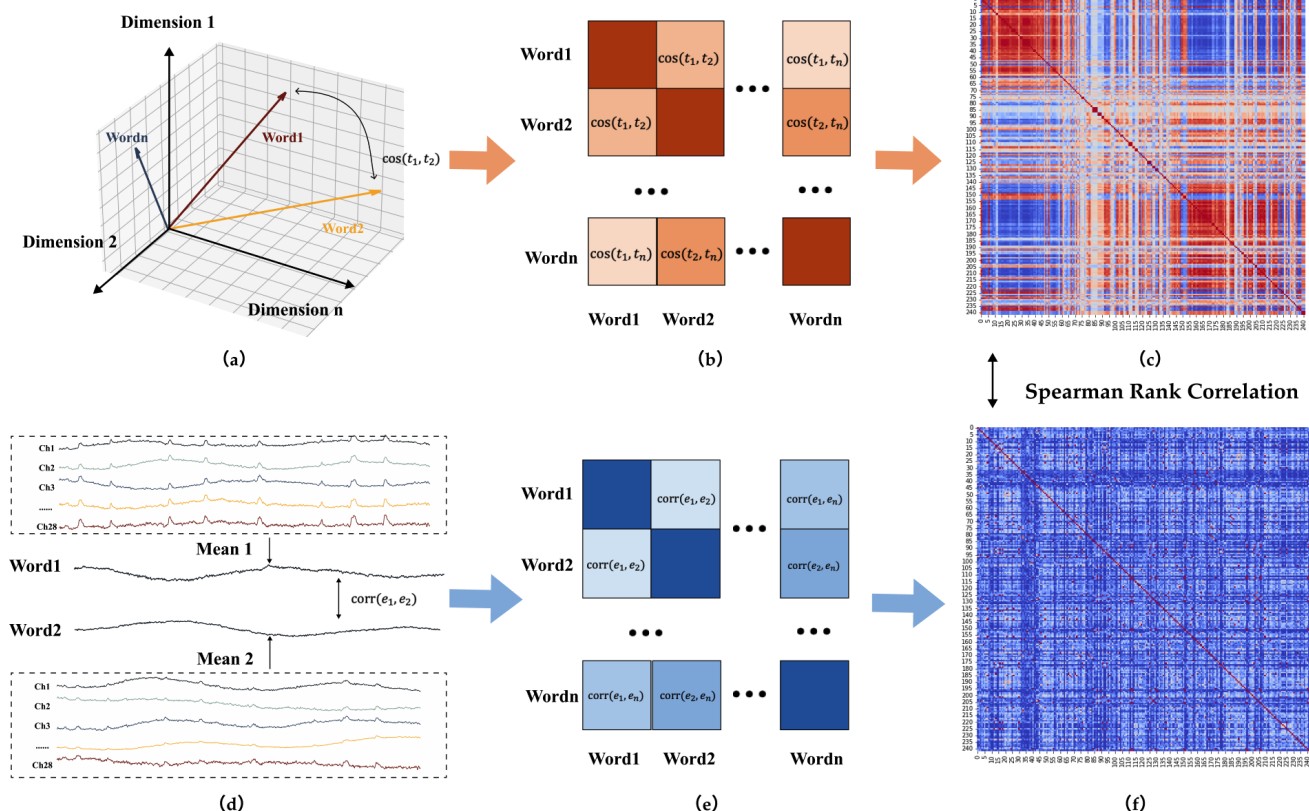

**Figure 3.** Steps for calculating representational similarity. (**a**) Compute the cosine similarity of word vectors from deep language models. (**b**) Arrange the cosine similarities of the word vectors into a matrix. (**c**) An example of the computed word vector similarity matrix. The deeper the blue, the lower the similarity; the deeper the red, the higher the similarity. (**d**) Compute the Pearson correlation coefficient for EEG vectors. (**e**) Arrange the Pearson correlation coefficients of the EEG vectors into a matrix. (**f**) An example of the computed EEG vector similarity matrix. Compare it with the corresponding word vector similarity matrix using the Spearman rank correlation coefficient.

The first step was to compute the representational similarity matrix for the deep language model. Initially, the cosine similarity between the word vectors of the 1220 tokens in the article was calculated, as described in Equation (6), where $v(t_i)$ represented the word vector of the i-th token. These similarities were arranged into a matrix, with the calculation process illustrated in Figure 3a–c. This similarity matrix's upper and lower triangles were mirror images of each other, with its diagonal representing the similarity of a word to itself.

$$Cosine\ Similarity(t_i, t_j) = \frac{v(t_i) \cdot v(t_j)}{\|v(t_i)\|_2 \times \|v(t_j)\|_2} \tag{6}$$

The second step involves computing the representational similarity matrix for the EEG signals. Similarly, the Pearson correlation coefficients between the EEG vectors corresponding to the 1220 tokens in the article are calculated. In Equation (7), $e(t_{i,k})$ represented the EEG signal of the i-th token at the k-th sampling point, and $\bar{e}(t_i)$ denoted the average EEG signal of the i-th token, averaged across 375 sampling points. These similarities were then arranged into a matrix in the same order as in the first step, with the calculation process illustrated in Figure 3d–f.

$$Pearson's\ corr(e_i, e_j) = \frac{\sum_{k=1}^{375}(e(t_{i,k}) - \bar{e}(t_i))(e(t_{j,k}) - \bar{e}(t_j))}{\sqrt{\sum_{k=1}^{375}(e(t_{i,k}) - \bar{e}(t_i))^2}\sqrt{\sum_{k=1}^{375}(e(t_{j,k}) - \bar{e}(t_j))^2}} \tag{7}$$

The third step involved computing the Spearman rank correlation coefficient between the two matrices, as described in Equation (8), where $d_i^2$ represents the squared difference in ranks for each element between the two matrices, and n is the number of elements in the matrix. This correlation coefficient reflects the representational similarity between the model and human brain activities [30].

$$Spearman\ Rank\ Correlation = 1 - \frac{6\sum d_i^2}{n(n^2 - 1)} \tag{8}$$

In this study, we selected "GloVE" [36], "BERT" [37], and "GPT-2" [38] models to compare with human brain activity related to language comprehension. "GloVE" is a word embedding method based on global word frequency statistics and is considered a relatively traditional model. Both "BERT" and "GPT-2" are models based on the Transformer architecture, and they excel in handling long-distance dependencies and capturing complex structures within sentences. However, "BERT" is a bidirectional model primarily designed for understanding context, while "GPT-2" is a generative model often used for text generation. By comparing these three distinct types of models, we aimed to gain a more comprehensive understanding of the similarities and differences between models and the human brain in processing language.

## 4. Results

### 4.1. EEG Sensitivity to Different Fact-Extraction Methods

4.1.1. Based on Cosine Similarity

This section reports the inter-class similarities of the EEG vectors corresponding to the factual word groups and non-factual word groups obtained from four factual word extraction methods and the intra-class similarities for each group. The quantities of factual and non-factual words extracted using the four methods (dependency, entity, pos, and TF-IDF) are shown in Table 2.

**Table 2.** Quantities of factual and non-factual words.

| Extraction Methods | Number of Fact Words | Number of Non-Fact Words |
|---|---|---|
| Dependency | 522 | 698 |
| Entity | 250 | 970 |
| Pos | 563 | 657 |
| TF-IDF | 478 | 742 |

Based on the description in Section 3, each 1220 word was recorded with 1.5 s of EEG signal. After averaging across the 28 channels, each word corresponds to a 375-dimensional EEG vector. When computing the cosine similarity of EEG vectors for factual word groups and non-factual word groups obtained from different extraction methods, three distinct time segments of the 1.5-s EEG signal were analyzed for each extraction method. These three segments are as follows: (1) the entire 1.5-s duration, encompassing 375 sampling points, denoted as "Overall"; (2) the time window potentially capturing the N400 ERP component, ranging from 250 ms to 500 ms, which includes 62 sampling points, denoted as "N400"; and (3) the time window potentially capturing the P600 ERP component, ranging from 500 ms to 1000 ms, containing 125 sampling points, denoted as "P600".

Firstly, the inter-class cosine similarity between the EEG vectors of fact word groups and non-fact word groups obtained from the four extraction methods was computed, with results shown in Table 3.

**Table 3.** The inter-class similarity between the EEG vectors of factual and non-factual word groups.

| TOI | Dependency | Entity | Pos | TF-IDF |
|---|---|---|---|---|
| Overall | −0.90 | −0.99 | 0.07 | −0.52 |
| N400 | −0.94 | −1.0 | 0.88 | −0.45 |
| P600 | −0.94 | −1.0 | 0.89 | 0.44 |

Overall: 0–1500 ms; N400: 250–500 ms; P600: 500–1000 ms.

It can be observed that the inter-class cosine similarity between the EEG vectors of fact word groups and non-fact word groups extracted using the Entity and Dependency methods is close to −1. The result indicates that the EEG signals for these fact word groups and non-fact word groups, as identified by these two methods, are highly dissimilar in direction and exhibit clear differentiation. In contrast, the EEG signals' differentiation between the two groups of words extracted based on pos is not pronounced, with the vectors even displaying a degree of positive correlation. Furthermore, the two groups of words extracted using TF-IDF exhibited significant differences in the N400 (often indicative of semantic comprehension) and P600 (typically representing syntactic processing) time windows, suggesting that the TF-IDF method captures some semantic information (as evidenced by the negative similarity in the N400 time window) but overlooks crucial syntactic information (evidenced by the positive similarity in the P600 window).

Subsequently, the intra-class similarity for factual word groups obtained from the four extraction methods was computed, with the results shown in Table 4.

**Table 4.** The intra-class similarity of the EEG vectors of the factual word group.

| TOI | Dependency | Entity | Pos | TF-IDF |
|---|---|---|---|---|
| Overall | 0.0023 | −0.0007 | 0.0018 | 0.0150 |
| N400 | 0.0038 | −0.0022 | 0.0024 | 0.0175 |
| P600 | 0.0023 | −0.0060 | 0.0021 | 0.0144 |

Overall: 0–1500 ms; N400: 250–500 ms; P600: 500–1000 ms.

All extraction methods yielded a very low intra-class cosine similarity for the factual word group, nearing 0. The result indicates a minimal correlation between the EEG vectors corresponding to the factual words extracted by these four methods. The angles between the vectors are nearly orthogonal, reflecting the brain's highly independent interpretation of each word within the factual word group.

Lastly, the intra-class similarity for non-factual word groups obtained from the four extraction methods was computed, with the results presented in Table 5.

**Table 5.** The intra-class similarity for the EEG vectors of the non-factual word group.

| TOI | Dependency | Entity | Pos | TF-IDF |
|---|---|---|---|---|
| Overall | 0.0094 | 0.0140 | 0.0106 | 0.0021 |
| N400 | 0.0087 | 0.0140 | 0.0109 | 0.0020 |
| P600 | 0.0107 | 0.0156 | 0.0120 | 0.0031 |

Overall: 0–1500 ms; N400: 250–500 ms; P600: 500–1000 ms.

All extraction methods resulted in a very low intra-class cosine similarity for the non-factual word group, again approaching 0. The result indicates a minimal correlation between the EEG vectors corresponding to the non-factual words extracted by these four methods, and the brain's comprehension of each word within the non-factual word set is highly independent.

### 4.1.2. Based on EEG Signals

In order to visually illustrate the disparities in brain activity between factual and non-factual words acquired through four extraction methods (dependency, entity, pos, TF-IDF), corresponding EEG signal curves were graphed, as depicted in Figures 4–7. Figure 4

corresponds to the four extraction techniques. The curves represent the average EEG signals over 1.5 s for 14 participants reading all factual or non-factual words. The semi-transparent regions on either side of the curves indicate the voltage's standard error (SE) at corresponding time points. In the legend, "N" represents the number of EEG signals used for computing the average. For instance, in Figure 4, 7308 indicates that 14 participants read 522 factual words. For each extraction method, 4 subfigures display the voltage from different channels (Area of Interest, AOI). From left to right, these are all 28 channels (Overall), 10 channels located above the frontal lobe (Frontal), 10 channels above the central region (Central), and 8 channels above the parietal lobe (Parietal).

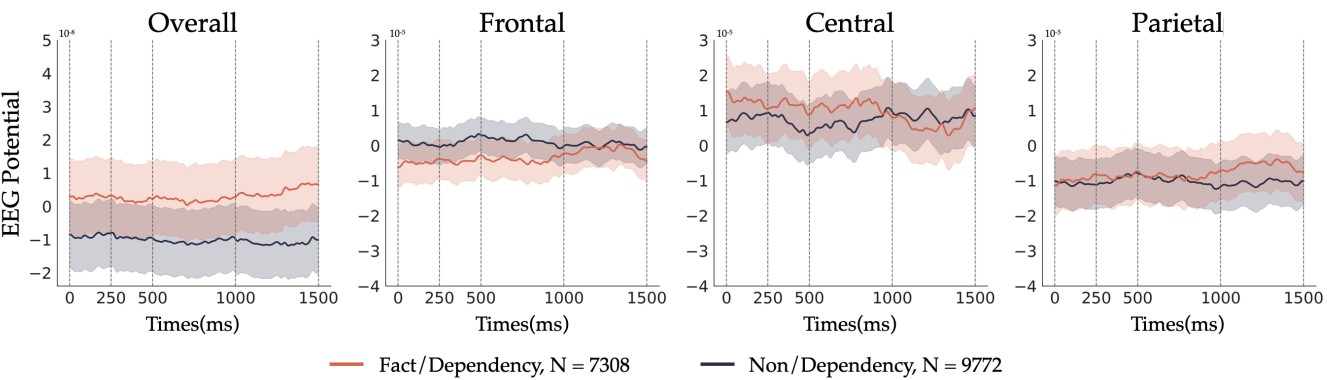

**Figure 4.** EEG signals obtained using the extraction method of dependency relations.

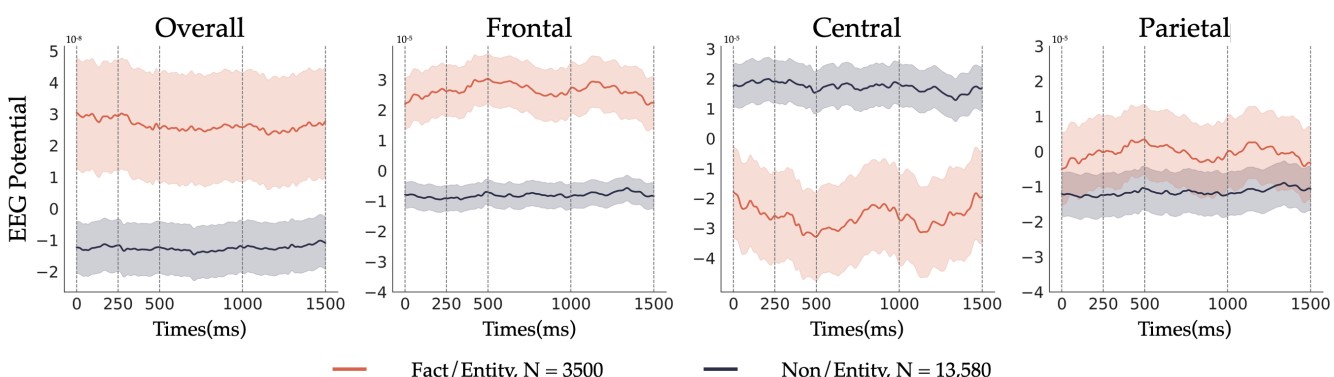

**Figure 5.** EEG signals obtained using the extraction method of named entity recognition.

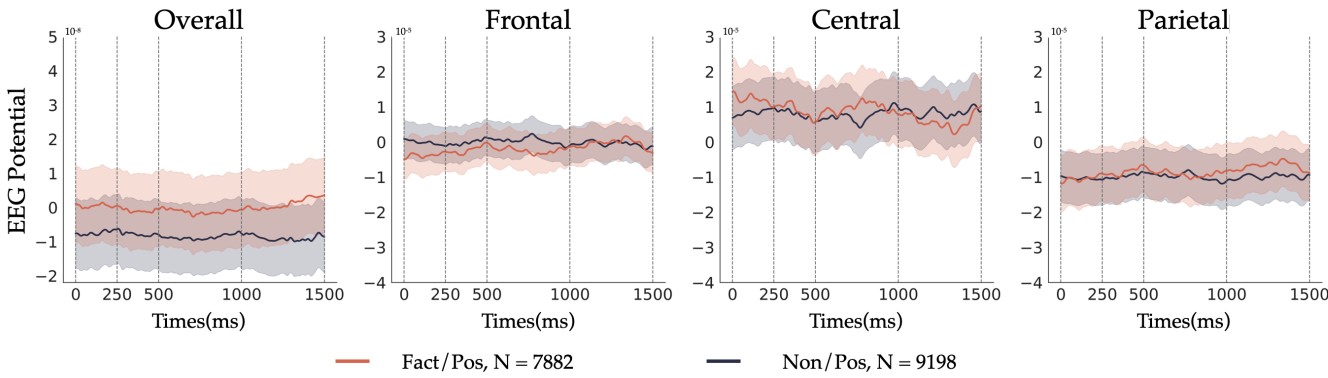

**Figure 6.** EEG signals obtained using the extraction method of part-of-speech tagging.

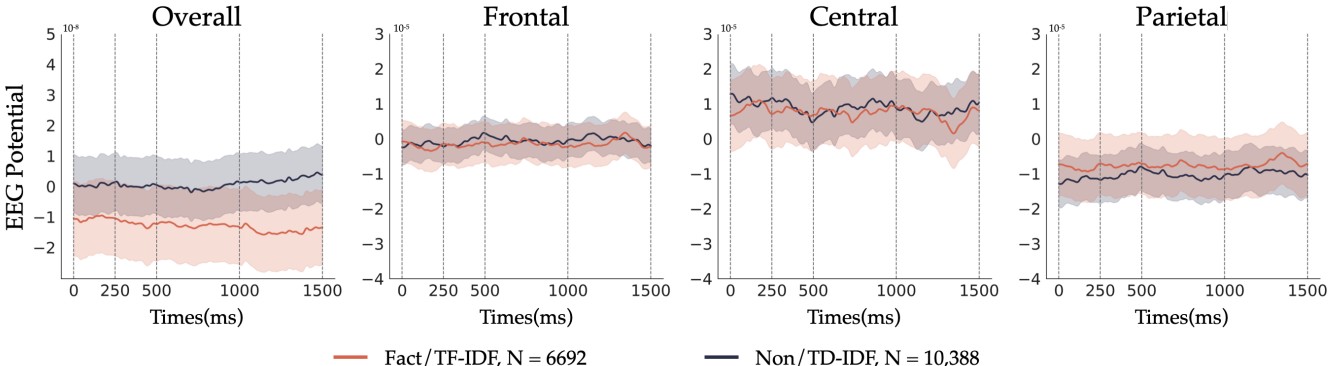

**Figure 7.** EEG signals obtained using the extraction method of TF-IDF.

Observation of the EEG signal curves reveals that it might be challenging to discern a clear distinction between the fact and non-factual word sets derived using the "Dependency", "Pos", and "TF-IDF" methods, as their voltage curves do not show significant differences. However, a marked contrast is observed with the curves obtained using the "Entity" method. In all four AOIs the EEG signal curves for factual and non-factual word sets identified through the "entity" method never intersect. This distinction is even more pronounced in the Overall, Frontal, and Central AOIs, where the signal curves, including their semi-transparent standard error margins, remain separate. This outcome aligns with previous findings based on cosine similarity. Additionally, research has shown that human brains exhibit discernible differences in electro-cortical manifestations when processing common and proper nouns during reading tasks [39]. Proper nouns are often extracted as factual words using the Entity method. These results further underscore that the distinction between groups of factual and non-factual words extracted through named entity recognition (entity) is most pronounced in the EEG signals.

### 4.1.3. Based on Global Field Power (GFP)

GFP values reflect the overall activity level of EEG signals within specific time windows [40]. Higher GFP values typically indicate stronger brain electrical activity, while lower values denote weaker activity. In this section, we calculated the GFP values of factual and non-factual words extracted using different methods across three time windows, as shown in Table 6.

**Table 6.** The GFP of different types of words.

| Type of Words | Overall | N400 | P600 |
|---|---|---|---|
| All words | 0.000528 | 0.000525 | 0.000524 |
| Dependency factual words | 0.000522 | 0.000520 | 0.000520 |
| Dependency non-factual words | 0.000532 | 0.000530 | 0.000527 |
| Entity factual words | 0.000542 | 0.000533 | 0.000538 |
| Entity non-factual words | 0.000524 | 0.000523 | 0.000520 |
| Pos factual words | 0.000525 | 0.000524 | 0.000523 |
| Pos non-factual words | 0.000530 | 0.000527 | 0.000525 |
| TF-IDF factual words | 0.000513 | 0.000510 | 0.000507 |
| TF-IDF non-factual words | 0.000537 | 0.000535 | 0.000535 |

Overall: 0–1500 ms; N400: 250–500 ms; P600: 500–1000 ms.

Throughout the entire time window, factual words extracted using the Entity method exhibited the highest GFP values. This indicates that the overall brain activity level is strongest when processing these types of words. The result might suggest that such words pose more significant cognitive challenges, requiring more cognitive resources for processing, or they may trigger more complex cognitive processing mechanisms in the

brain. The N400 time window (250–500 ms) is associated with semantic understanding. During this phase, non-factual words extracted via the TF-IDF method elicited the strongest electrical activity in the brain, suggesting that these words stimulate more brain activity during semantic processing than other types. Additionally, factual words extracted using the Entity method also showed the second highest GFP values, indicating that these words also activate the brain to a certain extent regarding semantic understanding. The P600 time window (500–1000 ms) is primarily related to syntactic processing. In this stage, factual words extracted using the Entity method again exhibited the highest GFP values, suggesting that these words might have particular importance or complexity in syntactic understanding, thereby inducing stronger brain electrical activity. These findings may reveal differences in cognitive processing among different types of words and how the brain dynamically adjusts its processing strategies based on the attributes of words, such as factuality. This information is invaluable for understanding the neural mechanisms of language processing.

### 4.2. Correlation between the Human Brain and Models

This research aims to examine whether the methods of extracting factual words (features), the models for generating word vectors (models), and the time windows of EEG signals (TOI) would significantly influence the representational similarity between human brain activity and model across different AOIs. Therefore, a three-way repeated measures Analysis of Variance (ANOVA) can be conducted for each AOI. The three independent variables encompass features, models, and TOIs. The features can be categorized into four types: "dependency", "entity", "pos", and "TF-IDF"; the models for generating word vectors can be divided into three types: "GloVe", "BERT", and "GPT-2"; and the time windows of the EEG signals are split into three categories: "0–1500 ms (Overall)", "250–500 ms (N400)", and "500–1000 ms (P600)". This study conducted detailed statistical analyses on the representational similarity obtained for each combination, aiming to ascertain if different Features, Models, and TOIs would influence the representational similarity between the human brain and the model.

#### 4.2.1. All 28 Channels (Overall)

The descriptive statistical results of the RSA scores for the human brain and model across all channels were presented in Table A1. Figure 8 depicted the distribution of representational similarity between human brain activity and language models at three different TOIs for various models and fact word extraction methods using boxplots. The rhombus symbols represent outliers, indicating data points that differ significantly from other observations. The error bars represent the 95% confidence interval (*CI*). At the 0–1500 ms TOI, the highest RSA score of 0.00302 was observed between the human brain and the GloVE model, using the TF-IDF method for fact word extraction. At the 250–500 ms TOI, the highest RSA score reached 0.00283 for the human brain and the GloVE model, achieved with the Pos fact word extraction method. For the 500–1000 ms TOI, employing the TF-IDF method for fact word extraction yielded the highest RSA score of 0.00309 with the BERT model.

Before conducting a repeated measure ANOVA on the RSA scores of the 28 channels, we first performed Mauchly's Sphericity Test. We reported the results of the sphericity test for the main effects and interactions that met the assumption of sphericity. For effects that violated this assumption, we applied the Greenhouse–Geisser correction. On these 28 channels, the main effects of 'Model', 'Feature', and 'TOI', as well as the interaction effect between 'Model' and 'Feature', met the assumption of sphericity. Thus, we conducted a repeated measures ANOVA using the original degrees of freedom. For other three-way interactions that did not meet the assumption, we used the Greenhouse–Geisser correction. The results of the main and interaction effects are shown in Table A5.

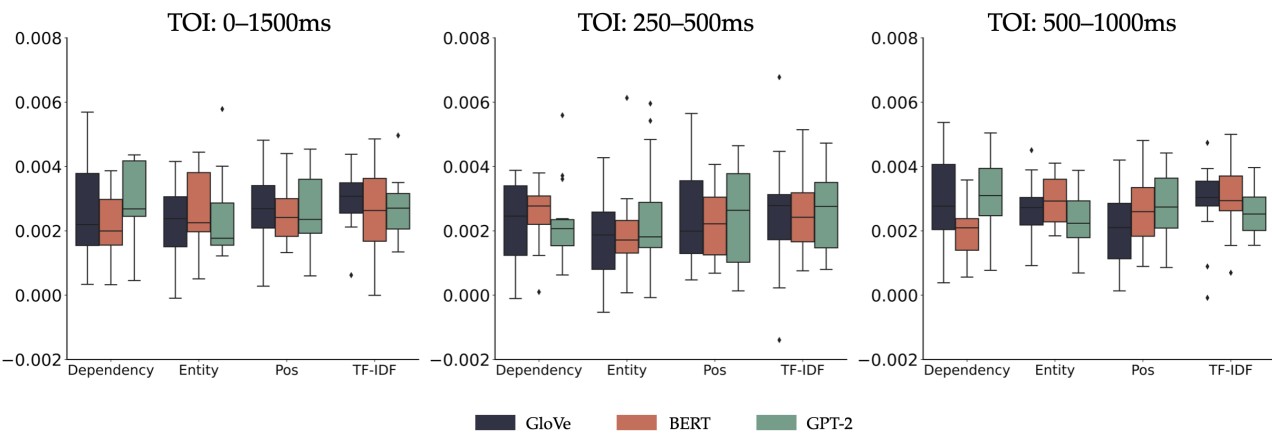

**Figure 8.** Distribution of RSA scores across all channels.

For the 28 channels, the main effect results were as follows: For the 3 models, the sphericity assumption was met ($p = 0.782$), with an ANOVA result of $F(2) = 0.334$, $p = 0.719$, indicating that the main effect of the chosen model categories on RSA scores was not significant. For the four extraction methods, the sphericity assumption was met ($p = 0.98$) with an ANOVA result of $F(3) = 0.701$, $p = 0.557$, suggesting that the main effect of the chosen fact word extraction methods on RSA scores was not significant. For the TOIs, the sphericity assumption was met ($p = 0.155$) with an ANOVA result of $F(2) = 0.334$, $p = 0.719$, indicating that the main effect of the chosen periods of interest on RSA scores was not significant.

The results for the interactions were as follows: For the interaction between the model and extraction method, the sphericity assumption was met ($p = 0.970$) with an ANOVA result of $F(6) = 0.68$, $p = 0.666$, suggesting that the interaction between the two factors was not significant. For the interaction between the models and TOIs, the sphericity assumption was not met ($p = 0.046$), so the Greenhouse–Geisser correction was applied, resulting in $F(2.338) = 0.224$, $p = 0.833$, indicating a non-significant interaction. For the interaction between the extraction method and TOI, the sphericity assumption was not met ($p = 0.044$), so the Greenhouse–Geisser correction was applied, resulting in $F(3.112) = 0.353$, $p = 0.794$, indicating a non-significant interaction. For the three-way interaction, the sphericity assumption was not met ($p < 0.001$). After applying the Greenhouse–Geisser correction, the result was $F(4.336) = 2.253$, $p = 0.070$, suggesting that the interaction among the three factors was insignificant.

### 4.2.2. Ten Channels Located above the Frontal Lobe (Frontal)

A total of 10 previously selected 28 EEG channels are located above the frontal lobe. The descriptive statistical results of the RSA scores for the brain and model based on these 10 EEG channels are presented in Table A2. Figure 9 depicts the distribution of representational similarity between human brain activity and language models at three different TOIs for various models and fact word extraction methods using boxplots. The rhombus symbols represent outliers, indicating data points that differ significantly from other observations. The error bars represent the 95% *CI*. At the 0–1500 ms TOI, the highest RSA score of 0.00435 was observed between the human brain and the GloVE model, using the TF-IDF method for fact word extraction. At the 250–500 ms TOI, the highest RSA score reached 0.00361 for the human brain and the GloVE model, achieved with the Dependency fact word extraction method. For the 500–1000 ms TOI, employing the Entity method for fact word extraction yielded the highest RSA score of 0.00401 with the GloVe model.

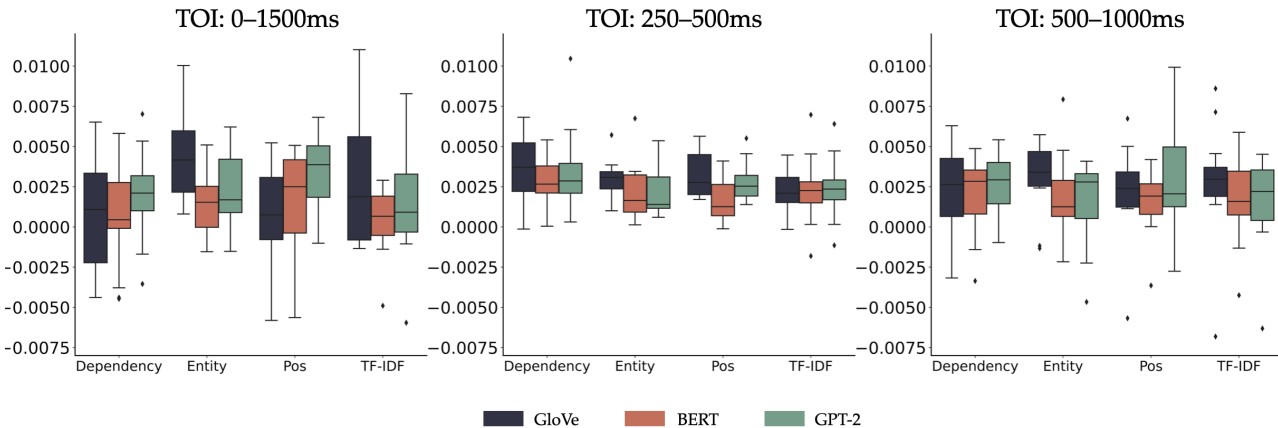

**Figure 9.** Distribution of RSA scores for channels located above the frontal lobe.

The results for the main and interaction effects from the repeated measures ANOVA are presented in Table A6. For the 10 channels above the frontal lobe, the main effect results were as follows: For the 3 models, the sphericity assumption was not met ($p = 0.022$). After applying the Greenhouse–Geisser correction, the result was $F(1.36) = 5.301$, $p = 0.025$, indicating a significant main effect of the selected three model categories on the RSA scores. The sphericity assumption was met for the four extraction methods ($p = 0.865$), resulting in $F(3) = 0.851$, $p = 0.475$, indicating that the main effect of the selected fact-word extraction methods on the RSA scores was not significant. For the TOI, the sphericity assumption was met ($p = 0.647$), resulting in $F(2) = 3.706$, $p = 0.038$, indicating a significant main effect of the selected TOIs on the RSA scores.

Interaction results were as follows: For the interaction between models and extraction methods, the sphericity assumption was met ($p = 0.073$) with $F(6) = 2.139$, $p = 0.058$, indicating no significant interaction. For the interaction between models and TOI, the sphericity assumption was not met ($p = 0.039$), and after applying the Greenhouse–Geisser correction, the result was $F(2.418) = 0.651$, $p = 0.556$, indicating no significant interaction. The sphericity assumption was met for the interaction between the extraction method and TOI ($p = 0.507$), resulting in $F(6) = 1.519$, $p = 0.183$, indicating no significant interaction. The sphericity assumption was not met for the interaction among the three factors ($p < 0.001$). After applying the Greenhouse–Geisser correction, the result was $F(5.054) = 1.081$, $p = 0.379$, indicating no significant interaction.

Given the significant main effects of the three models and three TOIs on the RSA scores, post-hoc tests were conducted on these two factors. The pairwise comparison results after the Bonferroni correction were presented in Tables 7 and 8.

**Table 7.** Post-hoc test for the main effects of the three models on RSA scores (frontal).

| Model 1 | Model 2 | Difference of $M$ (1–2) | $p$ | 95% CI | |
|---|---|---|---|---|---|
| | | | | Lower | Upper |
| GloVe | BERT | 0.001 | 0.050 | $-1.23 \times 10^{-6}$ | 0.002 |
| | GPT-2 | <0.001 | 0.384 | 0 | 0.001 |
| BERT | GloVe | $-0.001$ | 0.050 | $-0.002$ | $1.23 \times 10^{-6}$ |
| | GPT-2 | $-0.001$ | 0.226 | $-0.002$ | 0 |
| GPT-2 | BERT | 0.001 | 0.226 | 0 | 0.002 |
| | GloVe | <0.001 | 0.384 | $-0.001$ | 0 |

**Table 8.** Post-hoc test for the main effects of the three TOIs on RSA scores (frontal).

| TOI 1 | TOI 2 | Difference of *M* (1–2) | *p* | 95% *CI* | |
| --- | --- | --- | --- | --- | --- |
| | | | | Lower | Upper |
| Overall * | N400 * | −0.001 | 0.041 | −0.002 | −3.00 $\times 10^{-5}$ |
| | P600 | <0.001 | 0.772 | −0.001 | 0.001 |
| N400 * | Overall * | 0.001 | 0.041 | $3.00 \times 10^{-5}$ | 0.002 |
| | P600 | <0.001 | 0.448 | 0 | 0.001 |
| P600 | N400 | <0.001 | 0.448 | −0.001 | 0 |
| | Overall | <0.001 | 0.772 | −0.001 | 0.001 |

Overall: 0–1500 ms; N400: 250–500 ms; P600: 500–1000 ms. TOIs marked with the symbol '*' indicate a significant difference between them.

From the perspective of the 10 channels above the frontal lobe, although the ANOVA results indicated significant differences among the 3 models, the post-hoc test results suggested no significant differences between any 2 of the 3 models.

Regarding the 3 TOIs, post-hoc test results indicated that the RSA for the 0–1500 ms interval is significantly lower than the 250–500 ms interval (*p* = 0.041), but there was no significant difference compared to the 500–1000 ms interval (*p* = 0.772). Additionally, there was no significant difference between the RSAs for the 250–500 ms and 500–1000 ms intervals (*p* = 0.448).

### 4.2.3. Ten Channels Located above the Central Region (Central)

A total of 10 previously selected 28 EEG channels were located above the central region. The descriptive statistics for the RSA scores of the brain and the model on these 10 EEG channels were shown in Table A3. Figure 10 depicted the distribution of representational similarity between human brain activity and language models at three different TOIs for various models and fact word extraction methods using boxplots. The rhombus symbols represent outliers, indicating data points that differ significantly from other observations. The error bars represented the 95% *CI*. At the 0–1500 ms TOI, the highest RSA score of 0.00340 was observed between the human brain and the BERT model, using the Pos method for fact word extraction. At the 250–500 ms TOI, the highest RSA score reached 0.00346 for the human brain and the GPT-2 model, achieved with the Pos fact word extraction method. For the 500–1000 ms TOI, employing the TF-IDF method for fact word extraction yielded the highest RSA score of 0.00318 with the GloVe model.

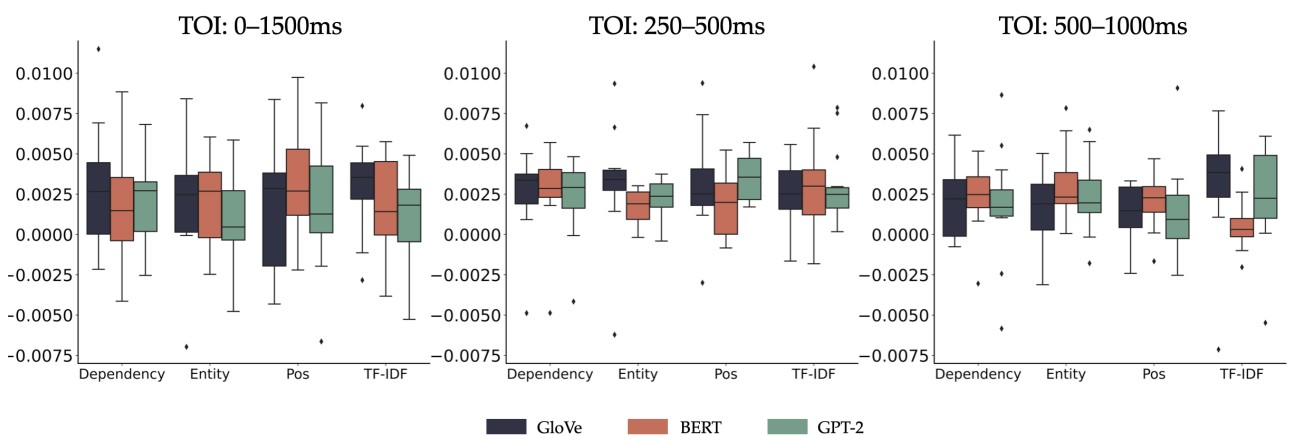

**Figure 10.** Distribution of RSA scores for channels located above the central region.

The results for the main and interaction effects from the repeated measures ANOVA were presented in Table A7. For the 10 channels located above the central region, the results for the main effects analysis were as follows: For the 3 models, the sphericity assump-

tion was met ($p = 0.321$), resulting in an ANOVA outcome of $F(2) = 0.048$, $p = 0.953$, indicating that the main effect of the chosen model categories on RSA scores was not significant. For the four extraction methods, the sphericity assumption was not met ($p = 0.014$), so the Greenhouse–Geisser correction was applied, resulting in a corrected ANOVA of $F(1.709) = 0.166$, $p = 0.816$, indicating that the main effect of the chosen fact word extraction methods on RSA scores was not significant. For the TOIs, the sphericity assumption was not met ($p = 0.031$), leading to a corrected ANOVA result of $F(1.389) = 4.140$, $p = 0.046$, suggesting a significant main effect of the chosen TOIs on RSA scores.

The interaction between the model and extraction method met the sphericity assumption ($p = 0.073$), with an ANOVA result of $F(6) = 0.68$, $p = 0.773$, indicating no significant interaction. For the interaction between the model and TOI, the sphericity assumption was not met ($p = 0.003$), so the Greenhouse–Geisser correction was applied, yielding a corrected ANOVA of $F(1.948) = 1.223$, $p = 0.310$, indicating no significant interaction. The sphericity assumption was violated for the interaction between the extraction method and TOI ($p = 0.005$), leading to a corrected ANOVA result of $F(3.588) = 0.395$, $p = 0.791$. For the three-way interaction, the sphericity assumption was not met ($p = 0.020$), resulting in a corrected ANOVA of $F(4.308) = 2.150$, $p = 0.082$. Given the significant main effect of the TOIs on RSA scores of the brain and model, post-hoc tests were conducted with Bonferroni correction, and the paired comparison results are shown in Table 9.

**Table 9.** Post-hoc test for main effects of the three TOIs on RSA scores (central).

| TOI 1 | TOI 2 | Difference of *M* (1–2) | *p* | 95% *CI* | |
|---|---|---|---|---|---|
| | | | | Lower | Upper |
| Overall | N400 | −0.001 | 0.165 | −0.003 | 0 |
| | P600 | <0.001 | 1 | −0.001 | 0.001 |
| N400 | Overall | 0.001 | 0.165 | 0 | 0.003 |
| | P600 | <0.001 | 0.131 | 0 | 0.002 |
| P600 | N400 | <0.001 | 0.131 | −0.002 | 0 |
| | Overall | <0.001 | 1 | −0.001 | 0.001 |

Overall: 0–1500 ms; N400: 250–500 ms; P600: 500–1000 ms.

For the three TOIs, pairwise differences were not significant. However, the RSA corresponding to N400 was higher than overall and P600.

### 4.2.4. Eight Channels Located above the Parietal Lobe (Parietal)

A total of 10 previously selected 28 EEG channels were located above the parietal lobe. The descriptive statistics for the RSA scores of the brain and the model on these 8 EEG channels are shown in Table A4. Figure 11 depicts the distribution of representational similarity between human brain activity and language models at three different TOIs for various models and fact word extraction methods using boxplots. The rhombus symbols represent outliers, indicating data points that differ significantly from other observations. The error bars represented the 95% *CI*. At the 0–1500 ms TOI, the highest RSA score of 0.00393 was observed between the human brain and the GloVE model, using the TF-IDF method for fact word extraction. At the 250–500 ms TOI, the highest RSA score reached 0.00308 for the human brain and the GPT-2 model, achieved with the Pos fact word extraction method. For the 500–1000 ms TOI, employing the Entity method for fact word extraction yielded the highest RSA score of 0.00407 with the GloVe model.

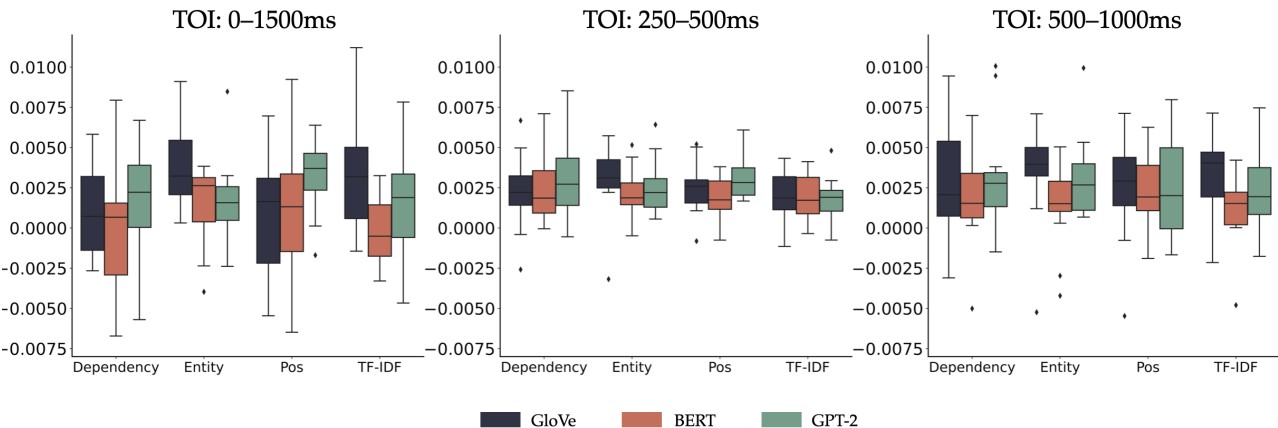

**Figure 11.** Distribution of RSA scores for channels located above the parietal lobe.

The results for the main and interaction effects from the repeated measures ANOVA are presented in Table A8. For the eight channels located above the parietal lobe, the main effects analysis results were as follows: For the three models, the sphericity assumption was not met ($p = 0.012$), leading to the application of the Greenhouse-Geisser correction, yielding a corrected ANOVA of $F(1.316) = 8.35$, $p = 0.007$, indicating a significant main effect of the model categories chosen on the RSA scores. For the four extraction methods, the sphericity assumption was not met ($p = 0.021$), necessitating the Greenhouse-Geisser correction, with the corrected ANOVA result being $F(1.875) = 0.415$, $p = 0.652$, implying the chosen fact word extraction methods did not have a significant main effect on RSA scores. The sphericity assumption was met for the TOIs ($p = 0.156$), resulting in an ANOVA outcome of $F(2) = 3.584$, $p = 0.042$, suggesting a significant main effect of the selected TOI on RSA scores.

The interaction between the model and extraction method did not meet the sphericity assumption ($p = 0.033$), leading to the Greenhouse-Geisser correction and a corrected ANOVA result of $F(3.468) = 2.012$, $p = 0.118$, indicating no significant interaction. The interaction between the model and TOI did not meet the sphericity assumption ($p = 0.027$), leading to a corrected ANOVA of $F(2.609) = 1.706$, $p = 0.189$ after applying the Greenhouse-Geisser correction. The interaction between the extraction method and TOI did not meet the sphericity assumption ($p = 0.043$), resulting in a corrected ANOVA of $F(3.382) = 1.066$, $p = 0.378$ after the Greenhouse-Geisser correction. For the three-way interaction, the sphericity assumption was met ($p = 0.051$), leading to an ANOVA result of $F(12) = 1.071$, $p = 0.388$, implying no significant interaction.

Given the significant main effects of the three models and TOIs on the RSA scores of the brain and model, post-hoc tests were conducted using the Bonferroni correction, with paired comparison results presented in Tables 10 and 11.

**Table 10.** Post-hoc test for the main effects of the three models on RSA scores (parietal).

| Model 1 | Model 2 | Difference of *M* (1–2) | *p* | 95% *CI* Lower | 95% *CI* Upper |
|---------|---------|------------------------|-----|-------|-------|
| GloVe | BERT | 0.001 | 0.054 | $-1.56 \times 10^{-5}$ | 0.002 |
|       | GPT-2 | <0.001 | 1 | −0.001 | 0 |
| BERT | GloVe | −0.001 | 0.054 | −0.002 | $1.56 \times 10^{-5}$ |
|      | GPT-2 | −0.001 | 0.009 | −0.002 | 0 |
| GPT-2 | BERT | 0.001 | 0.009 | 0 | 0.002 |
|       | GloVe | <0.001 | 1 | 0 | 0.001 |

**Table 11.** Post-hoc test for the main effects of the three TOIs on RSA scores (parietal).

| TOI 1 | TOI 2 | Difference of *M* (1–2) | *p* | 95% *CI* | |
| --- | --- | --- | --- | --- | --- |
| | | | | Lower | Upper |
| Overall * | N400 * | −0.001 | 0.072 | −0.001 | $5.22 \times 10^{-5}$ |
| | P600 | −0.001 | 0.191 | −0.002 | 0 |
| N400 * | Overall * | 0.001 | 0.072 | $-5.22 \times 10^{-5}$ | 0.001 |
| | P600 | <0.001 | 1 | −0.001 | 0.001 |
| P600 | N400 | <0.001 | 1 | −0.001 | 0.001 |
| | Overall | 0.001 | 0.191 | 0 | 0.002 |

Overall: 0–1500 ms; N400: 250–500 ms; P600: 500–1000 ms. TOIs marked with the symbol '*' indicate a significant difference between them.

From the perspective of the eight channels above the parietal lobe, post-hoc tests for the three models indicated that while there were no significant differences between the GloVe model and the BERT ($p = 0.054$) or GPT-2 models ($p = 1$), the RSA scores for the BERT model with the brain were significantly lower than those for the GPT-2 model with the brain ($p = 0.009$).

Regarding the three TOIs, pairwise differences were not significant.

### 4.2.5. A Brief Summary

Our study conducted a detailed analysis of representational similarity across four AOIs: Overall, Frontal, Central, and Parietal. Each AOI corresponded to a specific set of EEG channels. We evaluated the similarity in brain and model reading activities under varying conditions, encompassing four extraction methods, three models, and three TOIs, as detailed in Table 12. Cells in the table marked with an asterisk (*) signify significant main or interaction effects from the repeated measures ANOVA, with 'ns' indicating no significant effects.

**Table 12.** ANOVA results of RSA scores for different AOIs.

| Within-Subjects Effect | Overall (28) | Frontal (10) | Central (10) | Parietal (8) |
| --- | --- | --- | --- | --- |
| Model | ns | * | ns | GPT-2 > BERT ** |
| Feature | ns | ns | ns | ns |
| TOI * | ns | $TOI_2 > TOI_1$ * | * | * |
| Model × Feature | ns | ns | ns | ns |
| Model × TOI | ns | ns | ns | ns |
| Feature × TOI | ns | ns | ns | ns |
| Model × Feature × TOI | ns | ns | ns | ns |

ns: $p > 0.05$. *: $p < 0.05$. **: $p < 0.01$; $TOI_1$: 0–1500 ms, $TOI_2$: 250–500 ms.

Our findings demonstrated no significant interaction effects among extraction methods, model selection, and TOIs across all AOIs, suggesting that each factor independently influences RSA scores. This implied that each variable, such as extraction method, model selection, or TOI, uniquely contributed to the correlation between human brain activity and the models.

Regarding TOIs, different intervals corresponded to distinct brain linguistic processing stages. For example, the N400 was typically linked with semantic violations or unexpected words, with the 250–500 ms TOI often used to explore this time window. This suggested that this particular TOI encompassed the semantic interpretation process. Results from channels above the Frontal lobe indicated higher RSA scores during the TOI, including the N400, compared to the overall duration. This suggested that the models captured semantic processing-related information to some extent, showing heightened sensitivity to semantic data over other types, like syntax or background knowledge.

Regarding the channels above the Parietal lobe, RSA scores for GPT-2 were notably higher than those for BERT. This could imply that GPT-2's processing strategies resonated more with the linguistic processing patterns of this specific brain region. From a model perspective, unlike BERT's bidirectional masked language model approach, GPT-2's unidirectional autoregressive model might have more closely mirrored participants' sequential text reading pattern. They could predict the next word in a text but could not see it in advance, akin to GPT-2's processing style.

## 5. Discussion

The brain, recognized as the only system capable of comprehending language, has sparked increasing interest among researchers exploring the interpretability of deep language models. This quest has led to leveraging the brain's linguistic cognitive mechanisms to decode these models' "black box." As outlined by Arana et al. [41], approaches to studying the relationship between models and brain activity fall into three distinct categories. The first category entails recording human behavioral and brain activity data alongside model outputs during linguistic tasks. This approach facilitates direct comparison and analysis of the corresponding patterns between the human brain and the model. Notably, some researchers have utilized linear models to align GPT-2's activations with fMRI responses, offering robust evidence for the viability of applying cognitive neuroscience principles in model interpretation [42]. The second category revolves around specific derived metrics that extract features from human brain data and model outputs for quantitative analysis. A prime example of such metrics is comparing the model's "surprise" factor against brain activity. Originating from information theory, "surprise" measures the unexpectedness of a stimulus. Research by Heilbron et al. [43] demonstrated that the model's vocabulary "surprise" could effectively elucidate and predict brain responses, reinforcing that the brain is engaged in continuous probabilistic predictions. The third category contrasts the geometric representations obtained from human behavior or brain activity with those from model activity. The most prominent method in this category, used in our study, is RSA. Unlike seeking a one-to-one word representation correspondence between the model and the human brain, RSA assesses their representations' overall geometric structural similarities. This method identifies which aspects of a neural network model align closely with brain neural signals, thereby facilitating cross-validation between brain data and various computational models. Such an approach is instrumental in pinpointing which models most accurately reflect brain activity [30].

This study employed three different methods—word EEG vectors, EEG signal curves, and GFP—to explore the sensitivity of EEG signals to various factual word extraction approaches. Additionally, RSA was utilized to examine the similarity in text reading activities between humans and models under diverse conditions, including different fact word extraction methods, deep language models, and various human word reading time windows. Despite its innovation, the study acknowledges certain limitations. Firstly, the study's approach was somewhat indirect. It focused on four factual extraction methods but did not directly integrate EEG signals with ATS tasks, unlike what Hollenstein et al. did while collecting the ZuCo dataset [44]. Consequently, while our study offered a comparison of the effectiveness of ATS methods in the context of cognitive neuroscience, it necessitates further exploration to comprehensively understand the interplay between the human brain and text summarization methods. Secondly, the participants in this study were native Chinese speakers. Although they were selected based on their proficient English language skills and the materials were carefully chosen to minimize the use of rare words, it is undeniable that human processing of a first language differs fundamentally from that of a second language. In this respect, the distinction between human and model processing is incomparable. Lastly, although each word in this study was presented for a relatively extended period, the neural activity for the current word still depended, to some extent, on the words that were previously presented [45,46]. This interdependence is a consideration for this research and serves as a caveat for current findings.

In the subsequent research, we plan to employ different methods to correlate models with brain activity, specifically focusing on different factual extraction approaches and exploring their cognitive neural mechanisms. Studies have already correlated the 12 layers of the BERT model with the human language comprehension process, enhancing the interpretability of each layer's feature and improving the model's performance through fine-tuning [47]. In future work, we aim to delve deeper into the connections between models and human brain activity, using these insights to optimize the structure and function of models, particularly to enhance their performance in ATS. To better understand the correlation between the human brain and models, we plan to integrate other cognitive neuroscience research methods, such as Functional Magnetic Resonance Imaging (fMRI), functional Near-Infrared Spectroscopy (fNIRS), and eye-tracking technology closely related to text reading. These methods will provide more temporal information about brain activity. Through these studies, we hope to find a more precise approach to elucidate the relationship between the brain and deep language models, offering valuable contributions to integrating cognitive neuroscience and artificial intelligence.

## 6. Conclusions

Our conclusions can be summarized in four key points: Firstly, the inter-class cosine similarity analysis of EEG vectors revealed significant differences in brain signals between factual and non-factual word groups, especially when extracted using Entity and Dependency methods. Secondly, our analysis of EEG signal curves revealed a striking differentiation between factual and non-factual word groups, predominantly when employing the Named Entity Recognition (Entity) method. Thirdly, GFP analysis across time windows confirmed higher cognitive demand in processing factual words identified by the Entity method. Finally, our research extensively analyzed the representational similarity in brain and model reading activities across four AOIs—Overall, Frontal, Central, and Parietal—using various fact word extraction methods, language models, and TOIs. The findings showed that each factor independently influenced the RSA scores, as there were no significant interaction effects among these variables across all AOIs. In the Frontal lobe channels, higher RSA scores were observed in the N400 time window, associated with semantic processing, indicating the model's heightened sensitivity to semantic information. In the Parietal lobe channels, GPT-2 outperformed BERT, possibly due to GPT-2's unidirectional processing aligning more closely with the natural sequential reading patterns in the experiment. Overall, our findings offer innovative insights into brain-model correlations in language processing, highlighting the potential of EEG-based approaches to enhance NLP research.

**Author Contributions:** Conceptualization, L.L. and L.Z.; methodology, Y.Z. (Yingqi Zhu) and Z.Z.; software, Y.Z. (Yubo Zheng) and H.S.; validation, Y.L., Y.Z. (Yingqi Zhu) and L.D.; formal analysis, Z.Z.; investigation, Y.Z. (Yubo Zheng), Y.L. and L.D.; resources, L.L. and L.Z.; data curation, H.S. and S.G.; writing—original draft preparation, Z.Z.; writing—review and editing, L.L. and L.Z.; visualization, Z.Z.; supervision, Y.Z. (Yingqi Zhu); project administration, L.L.; funding acquisition, L.L. All authors have read and agreed to the published version of the manuscript.

**Funding:** This research was funded by National Natural Science Foundation of China (NSFC), No. 62176024 and Engineering Research Center of Information Networks, Ministry of Education.

**Institutional Review Board Statement:** The study was conducted according to the guidelines of the Declaration of Helsinki and approved by the Ethics Committee of the Beijing University of Posts and Telecommunications (Ethic approval code: 202302003).

**Informed Consent Statement:** Informed consent was obtained from all subjects involved in the study.

**Data Availability Statement:** The data can be found here: https://drive.google.com/open?id=1Qq9 P4ewmiSYDRxhUFPlHfDSZ5of6dZTt&usp=drive_fs (accessed on 2 November 2023).

**Acknowledgments:** We are immensely grateful to the Beijing University of Posts and Telecommunications students who participated in this work and provided us with their invaluable support.

**Conflicts of Interest:** The authors declare no conflict of interest. All authors have approved the manuscript and agreed with submission to this journal.

**Abbreviations**

The following abbreviations are used in this manuscript:

| | |
|---|---|
| NLP | Natural Language Processing |
| NLU | Natural Language Understanding |
| ATS | Automatic Text Summarization |
| ABS | Abstractive (Text) Summarization |
| EEG | Electroencephalography |
| GFP | Global Field Power |
| RSA | Representational Similarity Analysis |
| TOI | Time of Interest |
| AOI | Area of Interest |
| pos | part-of-speech |
| TF-IDF | Term Frequency-Inverse Document Frequency |

## Appendix A. Statistics of RSA Scores

*Appendix A.1. Descriptive Statistics of RSA Scores*

**Table A1.** Descriptive statistics of RSA scores across all channels.

| Features | Models | 0–1500 ms | | 250–500 ms | | 500–1000 ms | | N |
|---|---|---|---|---|---|---|---|---|
| | | *M* | *SD* | *M* | *SD* | *M* | *SD* | |
| Dependency | BERT | 0.00213 | 0.00102 | 0.00256 | 0.00098 | 0.00194 | 0.00083 | 14 |
| | GPT-2 | 0.00301 | 0.00121 | 0.00231 | 0.00126 | 0.00304 | 0.00133 | 14 |
| | GloVE | 0.00270 | 0.00163 | 0.00226 | 0.00134 | 0.00299 | 0.00147 | 14 |
| Entity | BERT | 0.00266 | 0.00124 | 0.00201 | 0.00141 | 0.00295 | 0.00078 | 14 |
| | GPT-2 | 0.00237 | 0.00128 | 0.00244 | 0.00178 | 0.00228 | 0.00095 | 14 |
| | GloVE | 0.00228 | 0.00124 | 0.00184 | 0.00142 | 0.00269 | 0.00095 | 14 |
| Pos | BERT | 0.00253 | 0.00091 | 0.00223 | 0.00108 | 0.00262 | 0.00113 | 14 |
| | GPT-2 | 0.00262 | 0.00109 | 0.00246 | 0.00160 | 0.00275 | 0.00114 | 14 |
| | GloVE | 0.00271 | 0.00117 | 0.00283 | 0.00218 | 0.00210 | 0.00125 | 14 |
| TF-IDF | BERT | 0.00271 | 0.00137 | 0.00250 | 0.00125 | 0.00309 | 0.00119 | 14 |
| | GPT-2 | 0.00270 | 0.00095 | 0.00263 | 0.00126 | 0.00256 | 0.00073 | 14 |
| | GloVE | 0.00302 | 0.00099 | 0.00252 | 0.00189 | 0.00288 | 0.00121 | 14 |

**Table A2.** Descriptive statistics of RSA scores for channels located above the frontal lobe.

| Features | Models | 0–1500 ms | | 250–500 ms | | 500–1000 ms | | N |
|---|---|---|---|---|---|---|---|---|
| | | *M* | *SD* | *M* | *SD* | *M* | *SD* | |
| Dependency | BERT | 0.00048 | 0.00311 | 0.00276 | 0.00142 | 0.00208 | 0.00243 | 14 |
| | GPT-2 | 0.00182 | 0.00279 | 0.00339 | 0.00253 | 0.00258 | 0.00197 | 14 |
| | GloVE | 0.00093 | 0.00334 | 0.00361 | 0.00211 | 0.00247 | 0.00261 | 14 |
| Entity | BERT | 0.00139 | 0.00189 | 0.00214 | 0.00173 | 0.00174 | 0.00260 | 14 |
| | GPT-2 | 0.00224 | 0.00250 | 0.00206 | 0.00144 | 0.00165 | 0.00254 | 14 |
| | GloVE | 0.00435 | 0.00252 | 0.00297 | 0.00116 | 0.00401 | 0.00439 | 14 |
| Pos | BERT | 0.00157 | 0.00322 | 0.00163 | 0.00135 | 0.00154 | 0.00188 | 14 |
| | GPT-2 | 0.00336 | 0.00255 | 0.00276 | 0.00116 | 0.00271 | 0.00324 | 14 |
| | GloVE | 0.00058 | 0.00307 | 0.00317 | 0.00134 | 0.00150 | 0.00416 | 14 |
| TF-IDF | BERT | 0.00047 | 0.00207 | 0.00234 | 0.00209 | 0.00164 | 0.00269 | 14 |
| | GPT-2 | 0.00148 | 0.00355 | 0.00233 | 0.00184 | 0.00159 | 0.00280 | 14 |
| | GloVE | 0.00259 | 0.00386 | 0.00235 | 0.00130 | 0.00278 | 0.00346 | 14 |

**Table A3.** Descriptive statistics of RSA scores for channels located above the central region.

| Features | Models | 0–1500 ms | | 250–500 ms | | 500–1000 ms | | N |
|---|---|---|---|---|---|---|---|---|
| | | M | SD | M | SD | M | SD | |
| Dependency | BERT | 0.00091 | 0.00526 | 0.00276 | 0.00248 | 0.00133 | 0.00479 | 14 |
| | GPT-2 | 0.00095 | 0.00512 | 0.00238 | 0.00232 | 0.00184 | 0.00335 | 14 |
| | GloVE | 0.00110 | 0.00803 | 0.00260 | 0.00266 | 0.00156 | 0.00599 | 14 |
| Entity | BERT | 0.00204 | 0.00264 | 0.00170 | 0.00105 | 0.00300 | 0.00219 | 14 |
| | GPT-2 | 0.00064 | 0.00293 | 0.00227 | 0.00126 | 0.00148 | 0.00406 | 14 |
| | GloVE | 0.00157 | 0.00498 | 0.00316 | 0.00336 | 0.00032 | 0.00617 | 14 |
| Pos | BERT | 0.00340 | 0.00327 | 0.00188 | 0.00209 | 0.00215 | 0.00176 | 14 |
| | GPT-2 | 0.00217 | 0.00496 | 0.00346 | 0.00140 | 0.00136 | 0.00287 | 14 |
| | GloVE | 0.00072 | 0.00696 | 0.00325 | 0.00305 | 0.00043 | 0.00459 | 14 |
| TF-IDF | BERT | 0.00175 | 0.00294 | 0.00302 | 0.00308 | 0.00049 | 0.00150 | 14 |
| | GPT-2 | 0.00108 | 0.00261 | 0.00296 | 0.00231 | 0.00237 | 0.00301 | 14 |
| | GloVE | 0.00263 | 0.00543 | 0.00270 | 0.00187 | 0.00318 | 0.00351 | 14 |

**Table A4.** Descriptive statistics of RSA scores for channels located above the parietal lobe.

| Features | Models | 0–1500 ms | | 250–500 ms | | 500–1000 ms | | N |
|---|---|---|---|---|---|---|---|---|
| | | M | SD | M | SD | M | SD | |
| Dependency | BERT | −0.00018 | 0.00397 | 0.00252 | 0.00211 | 0.00182 | 0.00275 | 14 |
| | GPT-2 | 0.00154 | 0.00363 | 0.00290 | 0.00243 | 0.00295 | 0.00334 | 14 |
| | GloVE | 0.00027 | 0.00418 | 0.00226 | 0.00224 | 0.00298 | 0.00341 | 14 |
| Entity | BERT | 0.00163 | 0.00238 | 0.00216 | 0.00148 | 0.00149 | 0.00258 | 14 |
| | GPT-2 | 0.00252 | 0.00460 | 0.00250 | 0.00167 | 0.00249 | 0.00400 | 14 |
| | GloVE | 0.00332 | 0.00490 | 0.00304 | 0.00211 | 0.00407 | 0.00397 | 14 |
| Pos | BERT | 0.00122 | 0.00397 | 0.00181 | 0.00143 | 0.00217 | 0.00217 | 14 |
| | GPT-2 | 0.00382 | 0.00332 | 0.00308 | 0.00128 | 0.00327 | 0.00459 | 14 |
| | GloVE | 0.00071 | 0.00361 | 0.00258 | 0.00162 | 0.00192 | 0.00434 | 14 |
| TF-IDF | BERT | −0.00034 | 0.00210 | 0.00191 | 0.00147 | 0.00123 | 0.00218 | 14 |
| | GPT-2 | 0.00178 | 0.00334 | 0.00180 | 0.00135 | 0.00237 | 0.00247 | 14 |
| | GloVE | 0.00393 | 0.00498 | 0.00200 | 0.00155 | 0.00338 | 0.00252 | 14 |

*Appendix A.2. ANOVA Results of RSA Scores*

**Table A5.** ANOVA results of RSA scores across all channels.

| Within-Subjects Effect | Mauchly's $p$ | Adjustment | $df$ | $\eta^2$ | F | $p$ |
|---|---|---|---|---|---|---|
| Model | 0.782 | - | 2 | $4.64 \times 10^{-7}$ | 0.334 | 0.719 |
| Feature | 0.980 | - | 3 | $2.51 \times 10^{-6}$ | 0.701 | 0.557 |
| TOI | 0.155 | - | 2 | $3.72 \times 10^{-6}$ | 1.353 | 0.276 |
| Model × Feature | 0.970 | - | 6 | $1.59 \times 10^{-6}$ | 0.680 | 0.666 |
| Model × TOI | 0.046 | Greenhouse-Geisser | 2.338 | $2.80 \times 10^{-7}$ | 0.224 | 0.833 |
| Feature × TOI | 0.044 | Greenhouse-Geisser | 3.112 | $1.11 \times 10^{-6}$ | 0.353 | 0.794 |
| Model × Feature × TOI | 0.001 | Greenhouse-Geisser | 4.336 | $5.12 \times 10^{-6}$ | 2.253 | 0.070 |

**Table A6.** ANOVA results of RSA scores for channels located above the frontal lobe.

| Within-Subjects Effect | Mauchly's $p$ | Adjustment | $df$ | $\eta^2$ | F | $p$ |
|---|---|---|---|---|---|---|
| Model * | 0.022 | Greenhouse-Geisser | 1.36 | $6.04 \times 10^{-5}$ | 5.301 | 0.025 |
| Feature | 0.865 | - | 3 | $7.05 \times 10^{-6}$ | 0.851 | 0.475 |
| TOI * | 0.647 | - | 2 | $3.06 \times 10^{-5}$ | 3.706 | 0.038 |
| Model × Feature | 0.080 | - | 6 | $1.81 \times 10^{-5}$ | 2.139 | 0.058 |
| Model × TOI | 0.039 | Greenhouse-Geisser | 2.418 | $6.02 \times 10^{-6}$ | 0.651 | 0.556 |
| Feature × TOI | 0.507 | - | 6 | $1.12 \times 10^{-5}$ | 1.519 | 0.183 |
| Model × Feature × TOI | 0.006 | Greenhouse-Geisser | 5.054 | $1.35 \times 10^{-5}$ | 1.081 | 0.379 |

*: $p < 0.05$.

**Table A7.** ANOVA results of RSA scores for channels located above the central region.

| Within-Subjects Effect | Mauchly's $p$ | Adjustment | $df$ | $\eta^2$ | $F$ | $p$ |
|---|---|---|---|---|---|---|
| Model | 0.321 | - | 2 | $7.33 \times 10^{-7}$ | 0.048 | 0.953 |
| Feature | 0.014 | Greenhouse-Geisser | 1.709 | $1.35 \times 10^{-5}$ | 0.166 | 0.816 |
| TOI * | 0.031 | Greenhouse-Geisser | 1.389 | $9.34 \times 10^{-5}$ | 4.140 | 0.046 |
| Model × Feature | 0.073 | - | 6 | $1.04 \times 10^{-5}$ | 0.773 | 0.593 |
| Model × TOI | 0.003 | Greenhouse-Geisser | 1.948 | $1.72 \times 10^{-5}$ | 1.223 | 0.310 |
| Feature × TOI | 0.005 | Greenhouse-Geisser | 3.588 | $6.85 \times 10^{-6}$ | 0.395 | 0.791 |
| Model × Feature × TOI | 0.020 | Greenhouse-Geisser | 4.308 | $3.45 \times 10^{-5}$ | 2.150 | 0.082 |

*: $p < 0.05$.

**Table A8.** ANOVA results of RSA scores for channels located above the parietal lobe.

| Within-Subjects Effect | Mauchly's $p$ | Adjustment | $df$ | $\eta^2$ | $F$ | $p$ |
|---|---|---|---|---|---|---|
| Model ** | 0.012 | Greenhouse-Geisser | 1.316 | 0 | 8.35 | 0.007 |
| Feature | 0.021 | Greenhouse-Geisser | 1.875 | $1.89 \times 10^{-5}$ | 0.415 | 0.652 |
| TOI * | 0.156 | - | 2 | $3.29 \times 10^{-5}$ | 3.584 | 0.042 |
| Model × Feature | 0.033 | Greenhouse-Geisser | 3.468 | $3.60 \times 10^{-5}$ | 2.012 | 0.118 |
| Model × TOI | 0.027 | Greenhouse-Geisser | 2.609 | $1.39 \times 10^{-5}$ | 1.706 | 0.189 |
| Feature × TOI | 0.043 | Greenhouse-Geisser | 3.382 | $1.92 \times 10^{-5}$ | 1.066 | 0.378 |
| Model × Feature × TOI | 0.051 | - | 12 | $6.32 \times 10^{-6}$ | 1.071 | 0.388 |

*: $p < 0.05$. **: $p < 0.01$.

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
