# Peer review of "Exploring the Cognitive Neural Basis of Factuality in Abstractive Text Summarization Models: Interpretable Insights from EEG Signals"

_applsci, doi:10.3390/app14020875_

Round 1
Reviewer 1 Report
Comments and Suggestions for Authors
The manuscript from Zhang and colleagues reported a study comparing EEG signal when processing factual vs. non-factual word tokens as classified via four different ATS methods. Based on inter-class similarity measures of EEG, they showed that the Named Entity methods most efficiently capture the difference between two categories of words both within the N400/P600 time windows and across the entire word segment. Additionally, they used RSA to suggest potentially different impacts of language models being used. They concluded that this study demonstrates the ATS models' neurobiological validity.
The manuscript is conceptually interesting. It covers an important niche in the literature that bridges NLP and cognitive neurophysiology, approaching a novel research question with valid methods. The manuscript is also well-written. However, I do spot several potential problems, which could be potentially improved, or if not, would need to be acknowledged as a limitation.
Major
-
All the ATS methods that the authors compared here are offline methods, however, the EEG signals are online. Therefore I wonder how much real insights a direct binding of the two would inform us about either how the human brain works or how these text abstraction methods work. I would expect that the authors have a better argument here. Or, maybe simply down-tuning the claim as something similar to “to prove/compare the neurobiological-validity of the ATS methods”, which would be still novel enough.
-
The introduction lacks a clear hypothesis regarding the expected effects. If this study is exploratory in nature, please also specifically claim this at the end of the introduction section.
-
Why do you test English language processing on a group of Chinese participants? Why not directly test with texts with Chinese stimuli? This creates a severe confound: namely the effet may merely capture a processing of factual vs. non-factual foreign language processing. Notably, first and second language processing is essentially different, this is irrespective of language proficiency.
-
Of all main analyses (inter/intra-class correlation, RSA etc), raw EEG voltage from all electrodes was averaged. This might be a bit too simplistic. For dimension reduction, one could go with either calculate the global field power (GFP), or average across typical electrodes for either the N400 or the P600 time windows.
Minor
-
There is good evidence that language models may not differ that much when neural model parameters are analyzed together with EEG or brain data (Sassenhagen et al., 2020; Schrimpf et al., 2021). These references could be discussed in the introduction, especially regarding the RSA results.
-
The presentation time of each word in this study is relatively long, but the neural activity for a word can still be somehow highly dependent on its preceding word(s) (overlap). In the cognitive neuroscience literature, this is typically dealt with by time-resolved regression (e.g., Broderick et al., 2018; Osorio et al., 2022), rather than treating all words as independent. This may need to be acknowledge as a caveat of the current study.
-
Quite a lot of the citations are not correct: e.g., line164, Chen et al., also line185, reference 28 (this is definitely not a good citation for the RSA approach). I would suggest the authors double check all citations.
References
Sassenhagen, J., & Fiebach, C. J. (2020). Traces of meaning itself: Encoding distributional word vectors in brain activity. Neurobiology of Language, 1(1), 54-76.
Schrimpf, M., Blank, I. A., Tuckute, G., Kauf, C., Hosseini, E. A., Kanwisher, N., ... & Fedorenko, E. (2021). The neural architecture of language: Integrative modeling converges on predictive processing. Proceedings of the National Academy of Sciences, 118(45), e2105646118.
Broderick, M. P., Anderson, A. J., Di Liberto, G. M., Crosse, M. J., & Lalor, E. C. (2018). Electrophysiological correlates of semantic dissimilarity reflect the comprehension of natural, narrative speech. Current Biology, 28(5), 803-809.
Osorio, S., Straube, B., Meyer, L., & He, Y. (2022). The role of co-speech gestures in retrieval and prediction during naturalistic multimodal narrative processing. PsyArXiv.
Comments on the Quality of English Languageno
Author Response
We sincerely appreciate your detailed and highly constructive comments, which has been instrumental in enhancing the quality and clarity of our paper. In response to your insightful comments, we have diligently corrected errors, addressed concerns, and made the necessary revisions to our manuscript. Please refer to the attached document for a careful response to each of your comments.

Reviewer 2 Report
Comments and Suggestions for Authors
The authors presented a new study for assessing the correlation among human EEG signals and fact extraction techniques. In addition, representational similarity analysis (RSA) was used to analyze the relationship between AI language models and brain activity. A dataset of EEG data from 14 human subjects was collected during a task of reading comprehension. Each subject was instructed to read a selected text presented in a computer screen word-by-word. The texts were carefully selected from the Frank’s corpus. Cosine similarity metric was computed for pairs of EEG signals corresponding to factual and non-factual word groups. Likewise, cosine similarity is computed for each pair of word vectors. For the analysis of the representational similarity between the human brain and AI models, a similarity matrix is constructed by computing the cosine similarity for each pair of word vectors and another for each pair of word vectors. Then the Spearman rank correlation coefficient was computed to measure the similarity between the two similarity matrices. One result is that TF-IDF does not align with cognitive patterns during natural reading. Experimental results from the channels above the parietal lobe indicate that the RSA scores for GPT-2 are significantly higher than those for BERT.
The approach taken by the authors is sound. The paper is well written and clear.
However, the following points must be addressed:
1. Captions of tables 2 to 8 are missing.
2. Typos and spelling errors annotated in the PDF file (applsci-2768097-peer-reviewed-v1.pdf) must be corrected.
3. The conclusions section does not reflect the main results of the study. It must also mention the main results for the fact extraction techniques and the models.

Comments on the Quality of English LanguageThe quality of English language is fine. Only minor corrections required.
Author Response

(The authors gave the same response as above.)

Reviewer 3 Report
Comments and Suggestions for Authors
1. This study compares the correlation between certain language models (GloVe, BERT, GPT-2) and human brain activity under certain physiological conditions using RSA. This comparison makes it possible to evaluate the differences with the aim of both a possible correction in these models and an in-depth analysis of the mechanisms underlying language processing in the human brain.
2. I think that the analysis carried out is original and innovative
3. The conducted comparative research is innovative and is a step both towards improving the current language models and towards understanding the neurophysiological basis of language coding in the human brain.
4. Improvement of the speech, the goal is to try to simplify the way of presenting the material, although the subject is very complex. In places it is clumsily written and difficult to read. Section material and methods are well presented. I have no recommendations to it.
5. The conclusion as I wrote in the review should be revised because it does not sound clear to the reader. Some of the analysis in it should go to the discussion section
6. The references are well chosen, but should be expanded a bit to better represent the controversy in the literature on the subject.
7. Tables and figures are very well illustrated. They can be supplemented with a little more explanation
Author Response

(The authors gave the same response as above.)

Round 2
Reviewer 1 Report
Comments and Suggestions for Authors
I thank the authors for carefully addressing my comments. I have no more reservations.